# FacialFlowNet: Advancing Facial Optical Flow Estimation with a Diverse Dataset and a Decomposed Model

### Jianzhi Lu*
Shanghai Key Laboratory of
Intelligent Information Processing,
School of Computer Science, Fudan
University
Shanghai, China
jzlu22@m.fudan.edu.cn

### Ruian He*
Shanghai Key Laboratory of
Intelligent Information Processing,
School of Computer Science, Fudan
University
Shanghai, China
rahe16@m.fudan.edu.cn

### Shili Zhou
Shanghai Key Laboratory of
Intelligent Information Processing,
School of Computer Science, Fudan
University
Shanghai, China
slzhou19@m.fudan.edu.cn

### Weimin Tan†
Shanghai Key Laboratory of
Intelligent Information Processing,
School of Computer Science, Fudan
University
Shanghai, China
wmtan@fudan.edu.cn

### Bo Yan†
Shanghai Key Laboratory of
Intelligent Information Processing,
School of Computer Science, Fudan
University
Shanghai, China
byan@fudan.edu.cn

## Abstract

Facial movements play a crucial role in conveying altitude and intentions, and facial optical flow provides a dynamic and detailed representation of it. However, the scarcity of datasets and a modern baseline hinders the progress in facial optical flow research. This paper proposes FacialFlowNet (FFN), a novel large-scale facial optical flow dataset, and the Decomposed Facial Flow Model (DecFlow), the first method capable of decomposing facial flow. FFN comprises 9,635 identities and 105,970 image pairs, offering unprecedented diversity for detailed facial and head motion analysis. DecFlow features a facial semantic-aware encoder and a decomposed flow decoder, excelling in accurately estimating and decomposing facial flow into head and expression components. Comprehensive experiments demonstrate that FFN significantly enhances the accuracy of facial flow estimation across various optical flow methods, achieving up to an 11% reduction in Endpoint Error (EPE) (from 3.91 to 3.48). Moreover, DecFlow, when coupled with FFN, outperforms existing methods in both synthetic and real-world scenarios, enhancing facial expression analysis. The decomposed expression flow achieves a substantial accuracy improvement of 18% (from 69.1% to 82.1%) in micro-expressions recognition. These contributions represent a significant advancement in facial motion analysis and optical flow estimation. Codes and datasets can be found here.

---
*These authors contributed equally to this work.
†Corresponding Authors.

## CCS Concepts

• **Computing methodologies** → **Computer vision**; *Computer graphics.*

## Keywords

facial optical flow dataset, decomposed facial flow, expression flow

**ACM Reference Format:**
Jianzhi Lu, Ruian He, Shili Zhou, Weimin Tan, and Bo Yan. 2024. FacialFlowNet: Advancing Facial Optical Flow Estimation with a Diverse Dataset and a Decomposed Model. In *Proceedings of the 32nd ACM International Conference on Multimedia (MM '24), October 28-November 1, 2024, Melbourne, VIC, Australia.* ACM, New York, NY, USA, 10 pages. https://doi.org/10.1145/3664647.3680921

## 1 Introduction

The human face could be the most encountered object in a person's life, which emphasizes the vital role of analyzing it [9, 22, 45, 48, 50]. Facial optical flow, representing facial movement, is crucial in applications like micro and macro expression recognition [26], facial motion capture [54], facial video generation [25, 52], and more. Despite its importance, the absence of a dedicated dataset and baseline has hindered its advancement.

The primary challenges in facial optical flow estimation involve: 1) Facial expressions arise from facial muscle movements, presenting a non-rigid motion [24] distinct from the rigid motion observed in general datasets like Sintel [7] and KITTI [34]; 2) Delicate expression movements are often overshadowed by overall head motion, as illustrated by the mouth region in Fig. 1(c). While state-of-the-art optical flow methods [18, 21, 44] can estimate the entangled facial flow Fig. 1(b), they are not able to separately isolate these local movements, which are crucial for facial expression analysis.

In this paper, we aim to precisely estimate facial optical flow and decompose it into two components: **Head Flow**, which signifies the isolated rotation and movement of the human head, and **Expression Flow**, representing the transformation of local facial expressions.

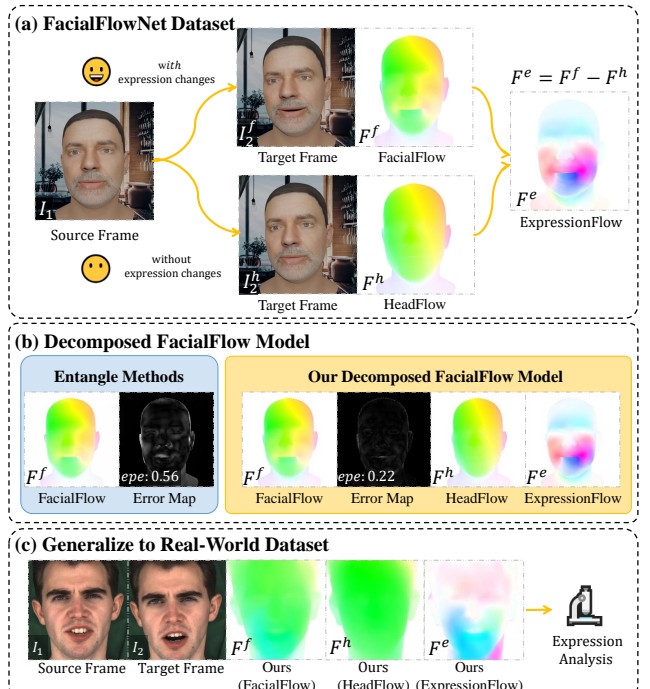

**Figure 1: The proposed FacialFlowNet dataset and DecFlow method. (a) FacialFlowNet (FFN) contains frames and optical flow labels with overall facial motion as well as head motion and expression. (b) DecFlow is designed to estimate accurate facial flow and further decompose it into head flow and expression flow. We show the optical flow and error map of GMA [21] (left) and our model (right). (c) Our method can generalize to real-world datasets like MEAD [46] and the expression flow can be utilized for downstream analysis.**

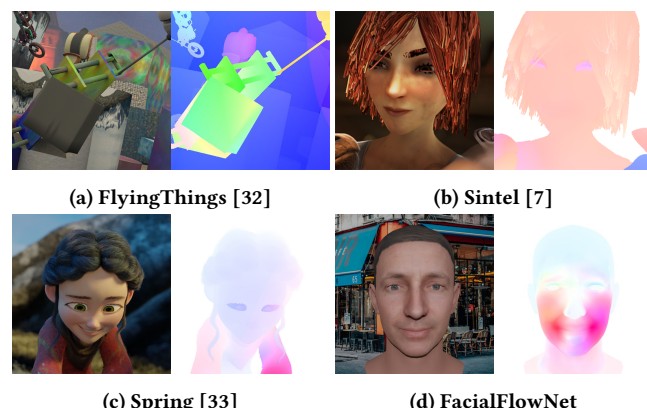

**Figure 2: Illustration of various optical flow datasets.**

expression flow achieves a substantial accuracy improvement of 18% (from 69.1% to 82.1%) in micro expressions recognition.

3) Extensive experiments demonstrate that FFN significantly enhances the accuracy of facial flow estimation across various optical flow methods, achieving up to 11% reduction in Endpoint Error (EPE) (from 3.91 to 3.48). Moreover, DecFlow outperforms other state-of-the-art methods, providing better insights for the analysis of facial movements.

## 2  Related Work

**General Optical Flow Estimation.** Optical flow estimation has been a fundamental vision task ever since this concept was brought by Berthold et al. [17]. Initially approached as an energy minimization problem, it utilized human-designed data and prior terms as optimization objectives [3–6, 51]. With the evolution of convolutional neural networks and deep learning, contemporary approaches adopt an end-to-end learning paradigm [13, 19, 42]. Notably, RAFT [44] represents a significant advancement in optical flow estimation, it constructs a 4D multiscale correlation volume and utilizes a GRU block to operate flows. Addressing the occlusion problem, GMA [21] and later works [18, 43] propose to use global feature and motion aggregation that could perceive long-range connections. These approaches primarily address general challenges such as fast-moving objects and occlusions. However, they are not designed to meet specific challenges like non-rigid motion and entangled representation in facial optical flow estimation, as mentioned earlier.

**General Optical flow datasets.** Datasets for optical flow estimation can be broadly categorized into real-world [15, 23, 34, 39, 40] data and synthetic data [7, 13, 14, 32, 33, 38, 41]. Among real-world data, KITTI [15, 34] is renowned in the domain of autonomous driving. It offers sophisticated training data derived from intricate device setups. On the synthetic side, Flyingchairs [13] and Flyingthings [32] generate optical flow labels by orchestrating random movements of foreground object models against a background. MPI Sintel [7], Monkaa [32] and Spring [33] gain the flow-image pairs from rendered animation movie scenes. AutoFlow [41] proposes an approach to search the hyperparameter for rendering training data, while RealFlow [16] synthesizes images using predicted optical

To achieve this, we propose **FacialFlowNet (FFN)**, a large-scale facial optical flow dataset. Our dataset, depicted in Fig. 1(a), is divided into two parallel segments: **FFN-F**, comprising images with overall facial motion and labels for facial flow, and **FFN-H**, offering corresponding frames and labels for head flow. By subtracting head flow from facial flow, we can isolate the expression flow.

Leveraging this dataset, we further propose **Decomposed Facial Flow (DecFlow)**, a novel optical flow estimation network featuring a facial semantic-aware encoder and a decomposed flow decoder. As illustrated in Fig. 1 (c), this approach can accurately estimate facial optical flow and further decompose it into head flow and expression flow, which can facilitate downstream facial analysis like dynamic facial expression recognition.

In summary, our main contributions are:

1) We contribute FacialFlowNet (FFN), a large-scale facial optical flow dataset comprising 105,970 pairs of realistic images from 9,635 identities, along with precise optical flow labels for both facial flow and head flow.

2) We present DecFlow, the first network capable of decomposing facial optical flow into head and expression flow. The decomposed

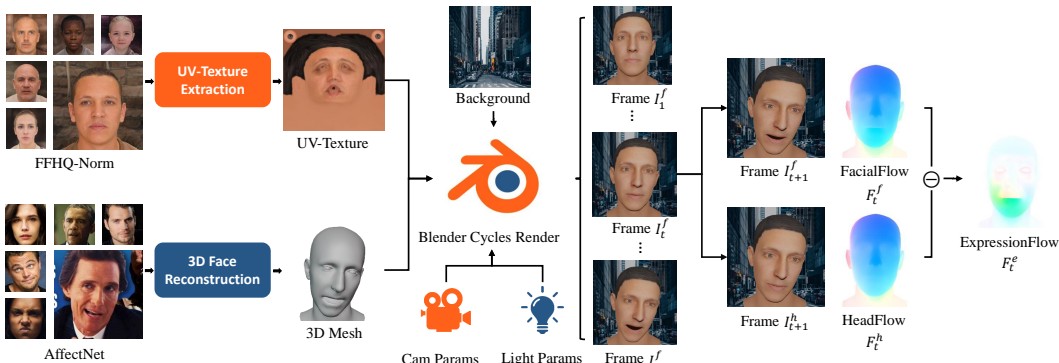

**Figure 3: The dataset generation pipeline. It takes a UV texture, a set of FLAME parameters, a background image, and camera/light parameters as input, producing video sequences of 5, 10, 15, or 20 frames with corresponding optical flow labels. $I_t^f$ and $I_t^h$ represent the $t^{th}$ frame in FFN-F and FFN-H respectively. From $I_t^f$ to $I_{t+1}^f$, we can get the facial flow, denoted as $F_t^f$. And from $I_t^f$ to $I_{t+1}^h$, we can obtain the head flow, indicated as $F_t^h$. Subtracting $F_t^h$ from $F_t^f$ results in the expression flow, denoted as $F_t^e$.**

flows to generate training data. However, as illustrated in Fig. 2, FlyingChairs [13] and FlyingThings [32] do not include any motion related to faces. Sintel [7] and Spring [33] do include frames with faces, but their character models have a cartoon style with facial features proportions differing significantly from real faces. Moreover, as observed in the optical flow visualization in Fig. 2, they treat the face as a rigid object, lacking the local non-rigid motion corresponding to real facial expressions. Therefore, these datasets are less suitable for predicting facial optical flow. In contrast, our dataset features realistic facial characteristics and simulates expressive movements.

**Facial Optical Flow Estimation.** While the above datasets and methods have achieved considerable success, the field of facial optical flow estimation has been overlooked. DeepFaceFlow [24] proposes a 3D facial flow dataset from 1,600 subjects and a corresponding network. Flow information is derived by calculating differences between face meshes obtained through 3D reconstruction based on images. However, disparities between meshes and actual human faces mean the flow in the dataset does not fully correspond to facial movements. Alkaddour et al. [1] presents a self-supervised approach and a dataset that consists of 41 subjects. They generate flows by partitioning faces into triangular meshes and calculating the local motion for each triangle. This partitioning may propose distinct artifacts in the ground truth of optical flow, making the labels unreliable. Unfortunately, the above datasets are not publicly available. Therefore, we contribute the FacialFlowNet dataset to accelerate the facial optical flow research with more diverse face images (9,635 identities) and more accurate optical flow labels with a rendering engine like Blender[1].

## 3  FacialFlowNet Dataset

We generate a realistic facial flow dataset with 9,635 unique faces, each displaying diverse shapes, expressions, and head poses. It comprises 105,970 image pairs at 512x512 pixels resolution, with corresponding flow labels. Similar to previous works [7, 13, 32, 33, 38],

we utilize Blender and its Cycles rendering engine for generating synthetic frames and flow labels. Our pipeline, illustrated in Fig. 3, takes a UV texture, a set of FLAME parameters, a background image, and camera/light parameters as input. It outputs video sequences with lengths of 5, 10, 15, or 20 frames, along with corresponding optical flow labels. The following sections provide details on the modules used for dataset generation.

### 3.1  3D Face Reconstruction

**Face model:** We use FLAME as our parameterized face model to reconstruct the meshes. FLAME [27] is a statistical model trained from around 33,000 3D face scans. It uses linear transformations to describe identity and expression-dependent shape variations, and standard linear blend skinning (LBS) to model neck, jaw, and eyeball rotations. It has parameters for identity shape $\beta \in \mathbb{R}^{|\beta|}$, facial expression $\psi \in \mathbb{R}^{|\psi|}$, and pose parameters $\theta \in \mathbb{R}^{3k+3}$ for rotations around $k = 4$ joints (neck, jaw, and eyeballs) and the global rotation. With all these parameters, FLAME can output a mesh with $n_v = 5023$ vertices. Formulated as:

$$M(\beta, \theta, \psi) \rightarrow (V, F) \tag{1}$$

with vertices $V \in \mathbb{R}^{n_v \times 3}$ and $n_f = 9976$ faces $F \in \mathbb{R}^{n_f \times 3}$.

**Face reconstruction:** We choose EMOCA [10] as our reconstruction method, this approach takes a single in-the-wild image and reconstructs a 3D face with sufficient facial expression detail to convey the emotional state of the input image. It enables us to generate high-quality face meshes with realistic expressions.

**Dataset:** For our dataset, which necessitates diversity in expressions, distinct identity faces, and high-quality images, we opt for AffectNet [35]. AffectNet stands out as one of the largest databases for facial expression, valence, and arousal. It contains more than 1M images with faces and extracted landmark points. We specifically select approximately a thousand images from each of the nine categories: neutral, happy, sad, surprise, fear, disgust, anger, contempt, and none, as the input for the reconstruction module.

**Expression sequence generation:** Give an input image $I$, we use EMOCA to obtain the FLAME parameters $(\beta_i, \theta_i, \psi_i)$. For generating

---

[1]https://www.blender.org

an expression sequence with $n$ frames, we initialize the parameters of the first frame as $(\beta_i, 0, 0)$ and the last frame's parameters as $(\beta_i, \theta_i, \psi_i)$, the remaining frames' parameters are obtained through interpolation. In FFN-F, the image pairs are composed of meshes from the $t$ and $t + 1$ frames, represented as $(M_t^f, M_{t+1}^f)$, where $t \in [1, n-1]$, they can be formulated as:

$$M_t^f = M\left(\beta_i, \frac{t\theta_i}{n}, \frac{t\psi_i}{n}\right) \qquad (2)$$

$$M_{t+1}^f = M\left(\beta_i, \frac{(t+1)\theta_i}{n}, \frac{(t+1)\psi_i}{n}\right) \qquad (3)$$

While in FFN-H, the image pairs consist $t^{\text{th}}$ mesh from FFH-F and the $t + 1^{\text{th}}$ mesh from its own, represented as $(M_t^f, M_{t+1}^h)$, which is formulated as:

$$M_{t+1}^h = M\left(\beta_i, \frac{(t+1)\,\theta_i}{n}, \frac{t\psi_i}{n}\right) \qquad (4)$$

Different from $M_{t+1}^f$, $M_{t+1}^h$ reserves the expression parameters from $M_t^f$ and only change it's pose parameters.

To simulate different rates of expression changes, we set $n$ to 5, 10, 15, and 20, to get different motion scales of 1/5, 1/10, 1/15, and 1/20. Through these steps, we can construct two corresponding datasets, each containing complete facial motion and isolated head motion, respectively.

## 3.2 UV-Texture Extraction

FLAME comes with an appearance model, which is converted from Basel Face Model's albedo space [37] to FLAME's UV layout [29]. To enhance the realism of our dataset, high-fidelity, and quality texture maps are essential. However, as depicted in Fig. 4(c), the existing method [28] yields low-quality, low-resolution textures, potentially resulting in unrecognizable and detail-lacking reconstructed results. AffectNet's in-the-wild variations further challenge texture quality. FFHQ-UV [2] offers a solution to this issue.

**Dataset:** FFHQ-UV [2] proposes a StyleGAN-Based Facial Image Editing module, creating FFHQ-Norm with consistent lighting, neutral expressions, and no occlusions from in-the-wild images. This dataset serves as the ideal input for our UV-texture Extraction module.

**Method:** FFHQ-UV's [2] textures are incompatible with FLAME, for they use the facial parametric model HiFi3D++ [8], which differs from FLAME in terms of both topology and vertex count. So we adapt their pipeline using a FLAME-based reconstruction method

and create a high-quality, FLAME-based UV-texture dataset. Refer to the supplementary materials for details.

## 3.3 Dataset Rendering

**Image Generation:** For image generation, we use the open-source 3D creation suite Blender. Inspired by [20, 32, 38], we aim to improve the network's robustness by integrating real photos as backgrounds in our dataset. We randomly select 400 background images from the internet, crop them to the same size, and ensure they encompass various indoor and outdoor scenes. During rendering, we position the image as a stationary plane behind the head model, manually adjusting the camera and lighting parameters to align the rendered results with the source images.

**Ground Truth Generation:** Modifying Blender's internal render engine pipeline enables the passage of vectors between different frames. The render pass is typically used for producing motion blur [7, 13, 32, 33, 38], and it produces the motion in the image space of each pixel; i.e., the ground truth of optical flow. Additionally, we also employ it to generate ground truth depth information, examples are available in the supplementary material.

## 3.4 Data Diversity

**Facial Expression Diversity:** We expect our dataset to inherit AffectNet's expression diversity. [35]. To confirm this, we employ DAN [47], a facial expression recognition network, to classify the 9000 faces selected from AffectNet [35] and our dataset. Fig. 5 visually presents the classification results. It's important to note that the classification model used is pre-trained on AffectNet, explaining the balanced distribution of the eight classification results in Fig. 5(a). Fig. 5(b) demonstrates that, despite a domain gap between synthetic and original images, our dataset effectively preserves a considerable degree of expression diversity.

**UV-Texture Diversity:** To assess identity diversity in our UV-Textures dataset, we calculate the identity vector using Arcface [11] for each image. The standard deviation and coefficient of variation of these vectors measure the identity variations. Tab. 1 presents evaluation results of the original dataset (FFHQ-Norm), and three rendered datasets with different UV-textures (FFHQ-UV [2], PO. [28] and ours). It shows that our textures preserve the most identity variations in FFHQ-Norm, whereas PO. hardly preserves identity differences. Fig. 5 also illustrates the same conclusion. To analyze

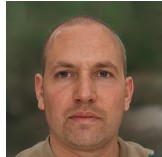 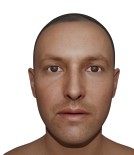 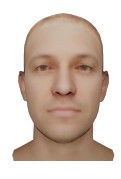 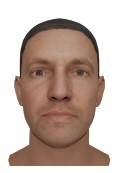

**(a) FFHQ-Norm**  **(b) FFHQ-UV**  **(c) PO.**  **(d) Ours**

**Figure 4: The rendered images with UV-Textures obtained from FFHQ-Norm with different methods.**

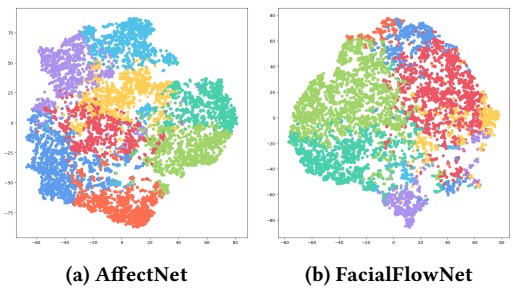

**(a) AffectNet**  **(b) FacialFlowNet**

**Figure 5: The t-SNE visualization of emotion features for both the AffectNet dataset and our dataset. Our dataset preserves a considerable degree of expression diversity.**

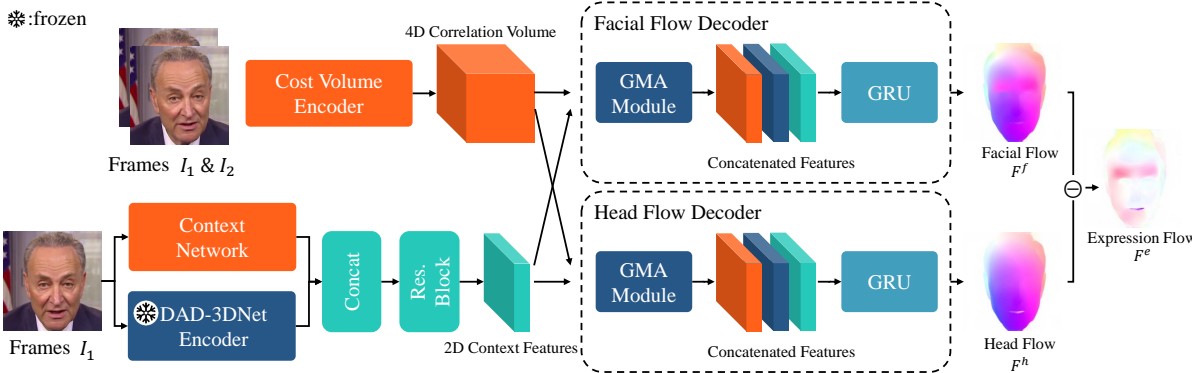

**Figure 6: The proposed DecFlow, featuring a facial semantic-aware encoder and decomposed decoder excels in accurately estimating and decomposing facial flow into head and expression components.**

**Table 1: Diversity evaluation of the proposed dataset in terms of identity feature standard deviation (Std.), coefficient of variation (Cv.), and identity cosine similarity scores (Sim.).**

| Metrics | FFHQ-UV [2] | PO. [28] | Ours |
|---|---|---|---|
| ID Std. ↑ | 0.023 | 0.021 | **0.029** |
| ID Cv. ↑ | 38.86 | 0.615 | **84.62** |
| ID Sim. ↑ | 0.721 | 0.620 | **0.783** |

the identity preservation from FFHQ-Norm to the rendered dataset, we also calculate the identity cosine similarity between each image in FFHQ-Norm and the rendered images. Tab. 1 indicates that the average identity similarity score of our dataset reaches 0.78, suggesting that our UV-textures can preserve the most facial features and details.

## 4 DecFlow Architecture

### 4.1 Overview

Our overall network diagram is shown in Fig. 6. We base our approach design on the successful recurrent architecture in RAFT [44] and GMA [21]. However, the previous architecture lacks the perception of facial semantic information and only provides an entangled optical flow for face movements. Therefore, we propose a facial semantic-aware encoder and a decomposed decoder to address the problem. Our model takes in a pair of consecutive frames of face movement and extracts the correlation volume and context feature. The decoder takes the context features and motion features as input, producing aggregated motion features that share information across the image. Finally, the two decoders predict the facial flow and head flow separately and obtain the expression flow by subtracting the head flow from the facial flow.

### 4.2 Facial Semantic-aware Encoder

Inspired by the extra encoders in SAMFlow [53] and MatchFlow [12], we posit that incorporating features with facial semantics can enhance the accuracy of facial optical flow prediction, given the

relatively fixed structure of the human face (nose, eyes, and mouth). To extract the 2D context features, we start by individually using the encoder of DAD-3DNet [31] and the context network of GMA [21] to encode the first image. The results from these encoders are concatenated and passed through a residual convolutional block to reduce channels and fuse features. Mathematically, this process can be expressed as:

$$\Phi_c = Res\left(E_G\left(I_1\right) \oplus E_D\left(I_1\right)\right) \tag{5}$$

where $E_G$ and $E_D$ are the GMA context network and DAD-3DNet encoder. The symbol $\oplus$ denotes the concatenation operator, and $Res$ corresponds to the residual convolutional block. During training, we freeze the parameters of DAD-3DNet's encoder and utilize only the pre-trained weights. This approach allows us to obtain context features with facial semantic information. In parallel, we employ GMA's cost volume encoder to extract 2D matching features and compute a 4D correlation volume.

### 4.3 Facial Decomposed Flow Decoder

We further propose a facial decomposed flow decoder that can decompose the facial motion into the head flow and expression flow. We adopted the same decoder structure as GMA, but the key difference is that we employed two parallel decoders to independently predict facial and head flow. The head flow $F^h$ is subtracted from the facial flow $F^f$ to obtain the expression flow $F^e$: $F^e = F^f - F^h$. In summary, our network takes two frames as input and outputs three types of flows: facial, head, and expression flow.

We first train the facial flow decoder and fix the parameters to train the head flow decoder. It is because the two decoders have separate optimization goals and can not simply co-train, which will cause a performance drop in facial flow (Tab. 5). When training the head flow decoder, we constrain head flow and expression flow as optimization targets. This way, the decoder can perceive the expression difference.

To supervise the estimating results, we define separate loss functions for facial, head, and expression flow. When training for the facial flow decoder, we use the optical flow loss in RAFT [44]. Then, we train the head flow decoder with the decomposed optical flow

**Table 2: Quantitative evaluation on FFN. The results include EPE on video sequences with a motion scale of 1/5, 1/10, 1/15, and 1/20. The metrics are the lower, the better. +FFN indicates finetuning on FFN. The best and second-best results are indicated in bold and underlined, respectively.**

| Methods | 1/20 | 1/15 | 1/10 | 1/5 | Sum. |
|---|---|---|---|---|---|
| RAFT [44] | 0.275 | 0.316 | 0.433 | 0.695 | 0.354 |
| GMA [21] | 0.267 | 0.306 | 0.422 | 0.664 | 0.343 |
| SKFlow [43] | 0.242 | 0.276 | 0.388 | 0.635 | 0.314 |
| FlowFormer [18] | 0.294 | 0.334 | 0.455 | 0.721 | 0.374 |
| RAFT+FFN | 0.111 | 0.130 | 0.182 | 0.316 | 0.148 |
| GMA+FFN | 0.108 | 0.125 | 0.172 | 0.296 | 0.142 |
| SKFlow+FFN | 0.112 | 0.133 | 0.189 | 0.331 | 0.152 |
| FlowFormer+FFN | 0.126 | 0.146 | 0.200 | 0.336 | 0.165 |
| DecFlow(Ours) | **0.098** | **0.116** | **0.162** | **0.284** | **0.132** |

loss $L = L_{head} + L_{expression}$, where $L_{head}$ and $L_{expression}$ are the optical flow loss [44] for head flow and expression flow.

## 5 Experiments

### 5.1 Implementation Details

We follow the standard optical flow training procedure [19, 42] of first training GMA [21] on FlyingChairs [13] for 120k iterations (batch size 8) and FlyingThings [32] for 120k iterations (batch size 6). Further training on MPI Sintel [7] for another 120k iterations (batch size 6). We then train our DecFlow on FacialFlowNet for 10k iterations (batch size 6) separately for the facial flow decoder and the head flow decoder, with the pre-trained weights from GMA. The training is performed on two 3090 GPUs with PyTorch [36], following GMA's hyperparameters and strategy [21].

### 5.2 Evaluation Datasets

**FacialFlowNet (FFN)** is divided into training, testing, and validation sets with a ratio of 97:2:1, comprising 90,193 pairs, 10,395 pairs, and 5,382 pairs of images respectively.

**MEAD** [46] contains high-resolution video frames with rich facial movements. From this dataset, we select 26 videos filmed from a frontal perspective, comprising a total of 3,402 frames. This subset is used to evaluate our method's efficacy in analyzing real facial dynamics. Additionally, we apply the method by Alkaddour et al. [1] to generate flow labels for 20,445 image pairs extracted from MEAD, denoted as **FMEAD**, for comparison with dedicated facial optical flow datasets.

**CK+** [30] has 593 acted facial expression sequences from 123 participants. We extracted the first and last frames of these sequences, resulting in 1,186 images for evaluating our method's performance in real facial optical flow estimation.

**CASME II** [49] contains 256 micro-expression videos from 26 subjects, featuring five prototypical expressions: happiness, disgust, repression, surprise, and others.

### 5.3 Quantitative Results on Synthetic Dataset

The primary evaluation metric is the average end-point error (EPE). To address the static background in FFN, we calculate the average

EPE within the moving regions using masks derived from depth information. Refer to the supplementary materials for more details. Note that, due to the relatively small facial motion magnitude, the differences between evaluation metrics are also small.

We divide the test set into four groups based on video length and evaluate various recent methods. The results are shown in Tab. 2. For fairness, we also finetune RAFT [44], SKFlow [43], FlowFormer [18], and GMA [21] on our training set. Our method achieves 7.0% higher accuracy than GMA, which indicates our approach can achieve superior accuracy while also being able to independently estimate facial flow, head flow, and expression flow.

### 5.4 Quantitative Results on Real World Datasets

We conduct experiments to validate our method's effectiveness in real facial flow estimation, addressing the domain gap between synthetic and real data. Due to the absence of ground-truth flow labels for real images, we compare the estimated flows with 3DMM's vertices and facial landmarks.

**Evaluation with 3DMM's vertices:** MICA [54] provide a metrical monocular tracker that can track facial motion by performing precise 3D face reconstruction on every frame of the video. Each reconstructed face mesh consists of 1787 facial vertices. We use it to perform facial motion tracking on images from MEAD [46] and CK+ [30], obtaining facial vertex coordinates for each image. As shown in Fig. 7, vertices were categorized into lips, forehead, cheeks, nose, and eye regions. For precision, we only used the 1435 vertices from the lips, cheeks, and eye regions, as the vertices in the nose and forehead regions remained relatively stationary.

To compare optical flow with the selected vertices, we use EPE as our metric. For each pair of images, such as $I_1$ and $I_2$, we obtain the facial vertex coordinates $C_1$, $C_2$, and the optical flow $Flow$. The average EPE was then calculated using the following formula:

$$E\left(I_1, I_2\right) = \sum_{i=1}^{n} \frac{F\left(\left(C_1^i - C_2^i\right), Flow\left(C_1^i\right)\right)}{n} \tag{6}$$

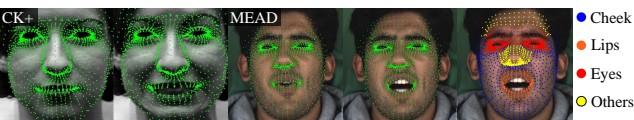

**Figure 7: The visualization of facial vertices on CK+ and MEAD. We partition the face into different regions, each displayed in various colors in this figure.**

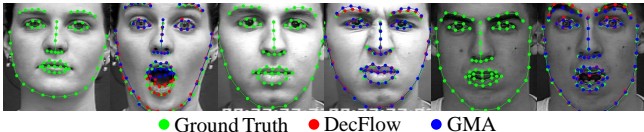

**Figure 8: Facial landmark labels provided by CK+. We compare the predicted optical flow with the coordinates of these landmarks.**

**Table 3: The quantitative evaluation results were obtained by comparing the EPE with landmarks or vertices on MEAD and CK+. (-) indicates the model is trained with standard procedure (C+T+S). FMEAD and FFN indicate the model is finetuned with FMEAD and FFN-F. (V.) and (L.) represent the coordinates of vertices and landmarks, respectively. ↓ signify performance improvement, while ↑ denote performance decline.**

| Methods | RAFT [44] | | | SKFlow [43] | | | FlowFormer [18] | | | GMA [21] | | | DecFlow(Ours) |
|---|---|---|---|---|---|---|---|---|---|---|---|---|---|
| | ( - ) | FMEAD | FFN | ( - ) | FMEAD | FFN | ( - ) | FMEAD | FFN | ( - ) | FMEAD | FFN | FFN |
| MEAD(V.) | 4.00 | 8.07 | 3.91 | 3.85 | 8.60 | 3.86 | 4.04 | 8.37 | 3.82 | 3.96 | 8.51 | **3.80** | **3.80** |
| | - | ↑4.07 | ↓0.09 | - | ↑4.75 | ↑0.01 | - | ↑4.33 | ↓0.22 | - | ↑4.55 | ↓0.16 | |
| CK+(V.) | 4.80 | 8.27 | **4.64** | 4.97 | 8.70 | 4.69 | 4.96 | 9.19 | 4.67 | 4.91 | 8.55 | 4.73 | 4.67 |
| | - | ↑3.47 | ↓0.16 | - | ↑3.73 | ↓0.28 | - | ↑4.23 | ↓0.29 | - | ↑3.64 | ↓0.18 | |
| CK+(L.) | 3.76 | 7.83 | 3.57 | 3.95 | 8.09 | 3.53 | 3.91 | 8.69 | 3.48 | 3.86 | 8.00 | 3.59 | **3.47** |
| | - | ↑4.07 | ↓0.19 | - | ↑4.14 | ↓0.42 | - | ↑4.78 | ↓0.43 | - | ↑4.14 | ↓0.27 | |

where $Flow(C_1^i)$ represents the motion vector at the coordinate $C_1^i$. And $F(V_1, V_2)$ represents the function used to calculate the EPE between the motion vectors $V_1$ and $V_2$.

In Tab. 3, FFN-F enhances the accuracy of all methods in estimating facial optical flow, with FlowFormer achieving the greatest improvement, reducing the EPE by 5.8% (from 4.96 to 4.67). However, finetuning on FMEAD leads to performance declines for all models. That's because Alkaddour's [1] method heavily relies on precise facial landmark predictions and triangle transformations, resulting in noticeable artifacts and inaccuracies in the flow labels. These findings underscore the superiority of FFN over both general datasets [7, 13, 32] and dedicated facial dataset [1].

Furthermore, DecFlow, which incorporates the ability to decompose facial flow, achieves comparable or superior accuracy in real facial optical flow estimation. Overall, with FFN and DecFlow, we achieve an accuracy improvement of up to 4.9% (from 4.91 to 4.67) compared to our baseline method (GMA with C+T+S).

**Evaluation with facial landmarks:** Due to the potential discrepancies between the reconstructed vertices and real faces, we also utilize the facial landmarks labels provided by CK+, as shown in Fig. 8. Since these labels are manually annotated, the evaluation results would be more precise. With these labels, we calculate the EPE using Eq. (6). The results in Tab. 3 demonstrate that FFN significantly improves the accuracy of various methods, achieving a maximum reduction of 11.0% (3.91 to 3.48) EPE for FlowFormer. Compared to the baseline (GMA with C+T+S), we achieve a maximum improvement of 10.1% (3.86 to 3.47) in facial flow accuracy. Visual examples in Fig. 8 illustrate that our method, in contrast to GMA, accurately predicts movements in the eyebrow and lip regions, aligning more closely with the ground truth.

### 5.5 Micro Expressions Recognition

We choose micro-expressions recognition as the downstream task and employ MMNet [26] to assess the performance of various optical flow networks. Then, we estimate the optical flow of the onset frames and apex frames in the original, untrimmed, and unaligned videos. These flows serve as inputs for the main branch of MMNet to learn motion-pattern features.

Tab. 4 shows the performance comparison of different optical flow models in both 5-class and 3-class classification tasks. After finetuning on FFN-F, various methods showed different degrees of

**Table 4: The results of micro-expressions recognition. +FFN indicates finetuning on FFN. DecFlow(F) and DecFlow(E) represent facial flow and expression flow. The best and second-best results are indicated in bold and underlined.**

| Methods | CASME II | | | |
|---|---|---|---|---|
| | (5 classes) | | (3 classes) | |
| | Accuracy | F1-Score | Accuracy | F1-Score |
| RAFT [44] | 73.1 | 0.694 | 87.8 | 0.839 |
| GMA [21] | 69.1 | 0.627 | 84.6 | 0.777 |
| SKFlow [43] | 73.9 | 0.686 | 88.4 | 0.823 |
| FlowFormer [18] | 66.6 | 0.603 | 82.6 | 0.744 |
| RAFT+FFN | 78.8 ↑5.7 | 0.732 ↑0.038 | 90.3 ↑2.5 | 0.858 ↑0.019 |
| GMA+FFN | 76.4 ↑7.3 | 0.730 ↑0.103 | 89.1 ↑4.5 | 0.867 ↑0.090 |
| SKFlow+FFN | 78.0 ↑4.1 | 0.744 ↑0.058 | 89.1 ↑0.7 | 0.854 ↑0.031 |
| FlowFormer+FFN | 77.6 ↑11.0 | 0.761 ↑0.158 | 88.4 ↑5.8 | 0.855 ↑0.111 |
| DecFlow(F) | 76.0 | 0.727 | 92.3 | 0.891 |
| DecFlow(E) | **82.1** | **0.818** | **94.2** | **0.931** |

performance improvement, with the maximum enhancement being 11.0%/0.158 and 5.8%/0.111 (FlowFormer) in terms of accuracy/F1-score. And compared to GMA [21] with general datasets (C+T+S), our approach and dataset demonstrated an improvement of up to 13.0%/0.191 and 9.6%/0.154 of accuracy/F1-score in both 5-class and 3-class classification. It highlights the superiority of our approach in estimating facial optical flow and the significance of decomposed expression flow in analyzing facial movements.

### 5.6 Qualitative Results on Real-World Images

Qualitative evaluation results of various methods are presented in Fig. 9. The samples in rows 1-4 have small emotional movements, entirely obscured by head motion. However, by using our method, these masked expression flows can be clearly decomposed. The examples in rows 5-8 depict expressions with intense emotional features and noticeable movements. Consequently, in the facial flow, the motion regions of these expressions are more pronounced. Compared to other methods, our approach predicts more accurate facial flow, visualized as clearer facial features, rather than focusing solely on large motion regions. Furthermore, by observing the last

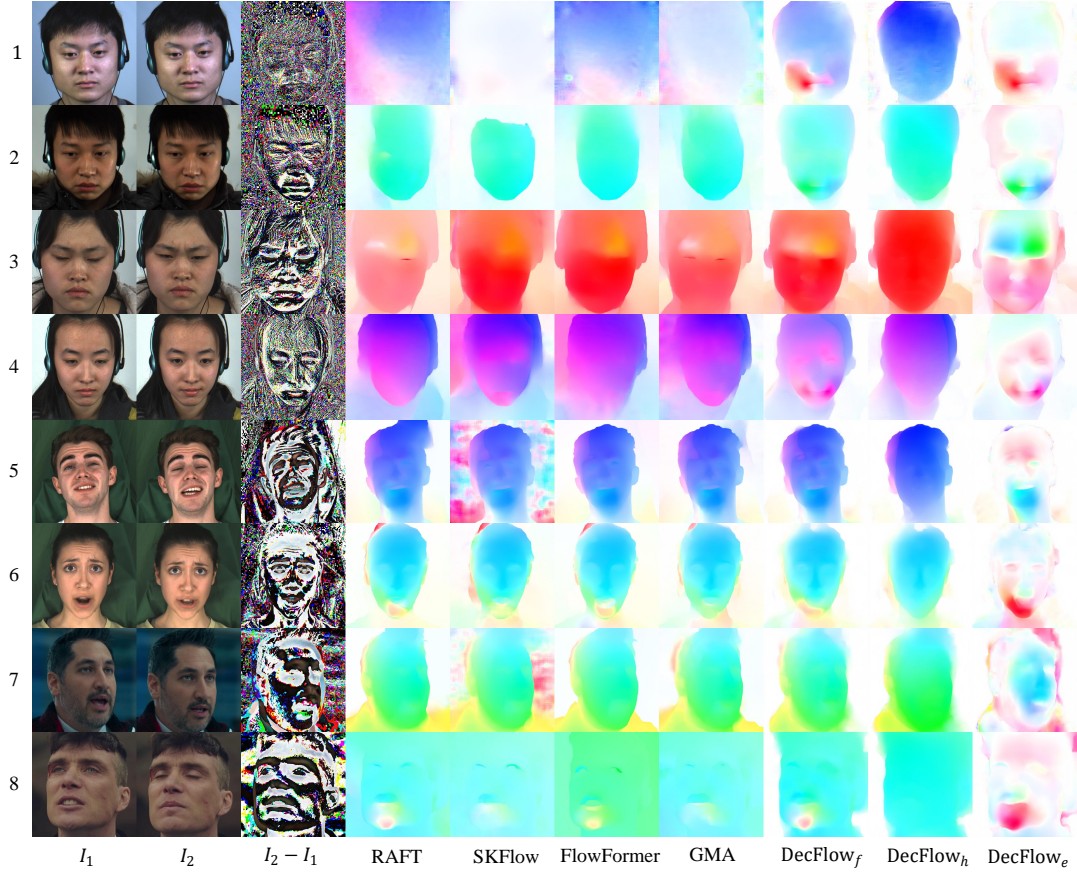

**Figure 9: Qualitative results on real-world images from CASME II [49] (1-4), MEAD [46] (5,6), and the Internet (7,8).** $DecFlow_f$, $DecFlow_h$, **and** $DecFlow_e$ **denote facial, head, and expression flow obtained by our method, respectively.**

column of Fig. 9, we can find that the decomposed expression flows normalize different facial movements to a similar range, visualized by similar colors. This information is crucial for facial expression analysis.

## 5.7 Ablation Study

Tab. 3 demonstrate that finetuning on FacialFlowNet enhances the accuracy of multiple baselines [18, 21, 43, 44] in both synthetic and real-world datasets, confirming the effectiveness of our dataset. Tab. 4 also highlights the significant advantage of expression flow over facial flow in micro-expressions recognition, confirming the effectiveness of our decomposed flow decoder. Subsequently, an ablation study validates our facial semantic-aware decoder, as presented in Tab. 5. Compared with GMA, adding another decoder for head flow estimation alone may affect the accuracy of facial flow. However, with the inclusion of the facial semantic-aware encoder, the network achieves superior accuracy compared to GMA while demonstrating the ability to decompose the facial flow.

## 6 Conclusion

This paper focuses on the challenges in non-rigid motion and entangled representation in facial flow estimation. We contribute

**Table 5: Ablation study of facial semantic-aware encoder.**

| Model | FFN | CK+(V.) | CK+(L.) |
|---|---|---|---|
| GMA [21] | 0.142 | 4.73 | 3.59 |
| DecFlow w/o Enc. | 0.145 | 4.69 | 3.52 |
| DecFlow (Ours) | **0.132** | **4.67** | **3.47** |

FacialFlowNet, a large-scale facial optical flow dataset with 9,635 identities and 105,970 image pairs. Our dataset significantly improves the accuracy of facial flow estimation across various optical flow methods. Additionally, we propose DecFlow, the first network capable of decomposing facial optical flow. Extensive experiments demonstrate the superior performance of our approach in facial flow estimation and expression analysis.

## 7 Acknowledgments

This work was supported in part by NSFC (No. U2001209, 62372117). The computations in this research were performed using the CFFF platform of Fudan University.

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
