# OpenReview forum: "FacialFlowNet: Advancing Facial Optical Flow Estimation with a Diverse Dataset and a Decomposed Model"
_acmmm.org/ACMMM/2024/Conference — MM2024 Poster_

### Official Review · Reviewer_tdV5 · 2024-05-23

**Rating:** 4
**Confidence:** 2

**Summary:**

The paper introduces FacialFlowNet (FFN), which is a novel large-scale facial optical flow dataset. Based on this dataset, the authors also propose the Decomposed Facial Flow Model (DecFlow), the first method capable of decomposing facial flow. Extensive experiments demonstrate the importance of FFN and the effectiveness of DecFlow.

**Strengths:**

- This work can bring contributions to multiple communities related to face analysis and face generation. Focused on faces, the proposed dataset provides precise optical flow labels for both facial flow and head movement. The proposed model has also been verified to achieve more accurate optical flow estimation compared to existing models.
- The paper is well-organized and easy to follow.
- The experiments are comprehensive, and the experimental results show that the proposed dataset and method can promote more accurate facial optical flow estimation.

**Limitations:**

- In the paper, the acquisition of GT comes from the render pass. I am curious whether it is possible to calculate optical flow using the vertices obtained from 3D Face Reconstruction. Specifically, since the vertices between two adjacent frames are directly corresponded, we can explicitly calculate the movement of each vertex, and then project it onto 2D to obtain facial optical flow. Compared with the method in this paper, the dataset obtained by this method can have real-world visual data instead of synthetic data.
- In general, the learning difficulty in a two-stage learning process is progressive. However, in the paper, the first stage requires the model to already be able to fully predict facial optical flow. Since expression flow can be derived by subtracting the head flow from the facial flow, the main task of the second stage is to predict the head flow, which is less challenging than the first stage. Moreover, the two decoders seem somewhat redundant; they should have some modules that can be shared. Streamlining the model structure could improve the model's efficiency.

**Suitability:**

3

---

### Official Review · Reviewer_hQ4P · 2024-05-24

**Rating:** 5
**Confidence:** 3

**Summary:**

This work aims to decouple the facial optical flow into head motion optical flow and facial expression optical flow. To achieve this goal, it starts by rendering a synthetic dataset, FacialFlowNet (FFN), using UV-texture extracted from FFHQ-Norm and facial expression extracted from AffectNet. Then, it builds a facial optical flow detection model, DecFlow, using RAFT and GMA, composed of two decoders designed to detect facial flow and head flow. Finally, the expression flow can be obtained by subtracting the head flow from the facial flow. Experiments on several benchmarks showcase the efficacy of DecFlow.

**Strengths:**

* Extensive ablation studies verify the effectiveness of different components of the dataset creation design and the model efficacy.
* Experiments on several benchmarks showcase the superiority of the introduced DecFlow
* The paper is well structured, and it is easy to follow.

**Limitations:**

* It is difficult to understand the main purpose of separating head flow from facial flow. I am curious about the main practical applications that require decoupling these two types of flows. Providing some figures to showcase a simple application would be much better. For example, is it possible / easier to perform expression editing with the decoupling?   How about enhancing the intensity of an expression, e.g., making a small smile into a big smile?

* It is unclear why the two decoders cannot be co-trained. L559 mentioned that it is because the two decoders have different objectives, and co-training them will lead to a performance drop. However, I do not understand which experiment in Tab. 5 showcases this conclusion. Besides, I am not convinced why it cannot be co-trained, given that each of them has a specific loss function to supervise. Does it suggest that combined training will lead to worse performance? It sounds counter-intuitive.

**Suitability:**

3

---

### Official Review · Reviewer_DxMK · 2024-05-26

**Rating:** 4
**Confidence:** 2

**Summary:**

The introduction of FacialFlowNet dataset comprising a large-scale collection of facial optical flow data with over 9,600 identities and 105,970 image pairs is a notable contribution to the field. The dataset addresses the scarcity of comprehensive facial optical flow datasets and provides a valuable resource for training and evaluation purposes.

**Strengths:**

The paper introduces DecFlow, a novel network designed to decompose facial optical flow, which is a significant contribution to the field.

**Limitations:**

The dataset would be beneficial for the paper to discuss in more detail the process of data collection, annotation, and potential biases present in the dataset to ensure its reliability and generalizability.

The evaluation of the proposed methodology could be strengthened by providing a more comprehensive comparison with existing state-of-the-art methods. Specifically, discussing how DecFlow performs compared to other facial flow estimation techniques in terms of accuracy, computational efficiency, and robustness across different datasets and scenarios would enhance the credibility of the proposed approach.

**Suitability:**

2

---

### Meta-Review · Area_Chair_XKB4 · 2024-07-02

**Recommendation:** Accept (Poster)
**Confidence:** 5

**Metareview:**

The manuscript proposes a face based optical flow method. The dataset and flow information is generated using Blender based rendering. A flow method is also presented and evaluation is performed on multiple relevant databases.

The reviewers indicate the novelty of the work. The rebuttal has clarified the reviewer queries.